# Cortical beta oscillations map to shared brain networks modulated by dopamine

Meera Chikermane[1]*, Liz Weerdmeester[1], Nanditha Rajamani[1], Richard M Köhler[1], Timon Merk[1], Jojo Vanhoecke[1], Andreas Horn[1,2,3], Wolf Julian Neumann[1,4]

[1]Department of Neurology, Movement Disorders and Neuromodulation Unit, Charité – Universitätsmedizin Berlin, Berlin, Germany; [2]Department of Neurology, Center for Brain Circuit Therapeutics, Brigham and Women's Hospital, Boston, United States; [3]Departments of Neurology and Neurosurgery, Massachusetts General Hospital, Boston, United States; [4]Einstein Center for Neurosciences Berlin, Humboldt Universitat, Berlin, Germany

## eLife Assessment

This study poses an **important** step forward in understanding the brain-network embedding of beta oscillations. The study advances our circuit-level understanding of the pathophysiology associated with dopaminergic alterations in psychiatric or neurological disorders. The study provides **compelling** evidence that beta oscillations across the neocortex and basal ganglia map onto shared functional and structural networks that show significant positive correlations with dopamine receptors.

*For correspondence:
meera.chikermane@charite.de

**Abstract** Brain rhythms can facilitate neural communication for the maintenance of brain function. Beta rhythms (13–35 Hz) have been proposed to serve multiple domains of human ability, including motor control, cognition, memory, and emotion, but the overarching organisational principles remain unknown. To uncover the circuit architecture of beta oscillations, we leverage normative brain data, analysing over 30 hr of invasive brain signals from 1772 channels from cortical areas in epilepsy patients, to demonstrate that beta is the most distributed cortical brain rhythm. Next, we identify a shared brain network from beta-dominant areas with deeper brain structures, like the basal ganglia, by mapping parametrised oscillatory peaks to whole-brain functional and structural MRI connectomes. Finally, we show that these networks share significant overlap with dopamine uptake as indicated by positron emission tomography. Our study suggests that beta oscillations emerge in cortico-subcortical brain networks that are modulated by dopamine. It provides the foundation for a unifying circuit-based conceptualisation of the functional role of beta activity beyond the motor domain and may inspire an extended investigation of beta activity as a feedback signal for closed-loop neurotherapies for dopaminergic disorders.

## Introduction

Oscillatory neural activity is proposed to orchestrate brain function by providing multilateral rhythmic synchronisation of neural excitability and action potential firing (*Fries, 2015*). One prominent example in the human brain is the beta rhythm (13–35 Hz), which has traditionally been described as a phenomenon of the motor domain, primarily localised to motor cortex and important for the pathophysiology of movement disorders, such as Parkinson's disease (PD) (*Hammond et al., 2007*; *Paulo et al., 2023*). In PD, beta activity, especially in deeper brain structures, such as the basal ganglia, has been associated with the loss of innervation with dopamine (*Iskhakova et al., 2021*), a neurotransmitter critically involved in synaptic plasticity and reinforcement learning at cortico-striatal synapses (*Berke, 2018*).

This relationship is mirrored in rodent and non-human primate models of PD, corroborating the role of dopamine as a key neurotransmitter governing beta synchrony. Meanwhile, the putative function of beta modulation in both human and animal literature has been extended within and beyond PD, to diverse domains of cognition (*Paulo et al., 2023*), memory (*Hanslmayr et al., 2016*), and emotions (*Alagapan et al., 2023*), but only a few studies so far have sought to relate these observations to dopamine signalling. For each of these domains, localised synchronisation or desynchronisation phenomena in the beta frequency range have been described in specific brain regions. What the common denominators of these brain regions are, and whether beta activity has a general structural and functional overlap with dopaminergic innervation remains unknown. This reflects a significant roadblock to a fundamental understanding of the circuit architecture of beta activity. To overcome this roadblock, we have developed a unique multimodal research data pipeline combining normative sources of invasive brain signal recordings, MRI connectomics, and positron emission tomography (PET). We address the hypothesis that beta activity is a ubiquitous synchronisation phenomenon that is characterised by whole-brain networks with subcortical nuclei that are under modulatory control of dopamine.

## Results

### Beta activity is a widely distributed resting brain rhythm in invasive cortical recordings

First, we aimed to identify the spatial distribution of cortical sources of beta activity in the resting human brain. Electrophysiology studies describing brain oscillations often rely on non-invasive electroencephalography (EEG) or magnetoencephalography (MEG) that provide access to brain signal recordings in healthy subjects. While beamforming and other source reconstruction techniques can provide estimates of the spatial peaks of certain brain activity patterns in these recordings, it can be hard to rule out that the derived brain regions are free from influence of volume conduction and current mixing from neighbouring brain areas (*Hedrich et al., 2017*). Thus, estimates of beta activity in prefrontal areas identified with EEG or MEG could to some extent be the result of motor cortex activity, contaminating sensors in frontal locations. Invasive recordings directly from brain tissue can circumvent such contamination, but are unavailable in healthy subjects. Here, we analysed over 30 hr of invasive resting-state brain signals from an open atlas (*Frauscher et al., 2018*) of the normal electroencephalogram in 1772 channels of either electrocorticography (ECoG; n = 258) or stereoelectroencephalography (sEEG; n = 1514) covering the human cortex. The atlas is derived from 106 epilepsy patients undergoing monitoring for seizure activity for respective surgery, but the signals were selected based on strict criteria to ensure their origin from healthy brain regions, in relevant distance to epileptogenic zones or lesions (*Frauscher et al., 2018*). The localised recordings from bipolar channels of one minute resting periods with eyes closed were taken to the frequency domain using Morlet wavelets and parametrised using the 'fitting oscillations and one over f' (FOOOF) toolbox (*Donoghue et al., 2020*), to extract the maximum peak amplitude of the periodic component of beta activity (13–30 Hz). For comparison and as a control, we extended this analysis to other canonical brain rhythms, including theta (4–8 Hz), alpha (8–12 Hz), and gamma (30–100 Hz) frequency bands. The frequency band with the highest peak amplitude was identified using the extracted peak parameter (pw) for each channel and depicted as the dominant rhythm for the respective localisation (*Figure 1*). In case of multiple peaks within the same frequency band, we focused the analysis on the peak with the highest amplitude. This revealed that 15.6% (277) channels were identified as theta-dominant, 22.4% (397) as alpha-dominant, 56.6% (1005) as beta-dominant, and 5.2% (93) as gamma-dominant channels. The focus of this article is beta oscillations, so all further analyses will refer to beta with alpha for comparison as it was the second most frequently occurring frequency band after beta. It should be noted that that alpha recordings were performed in eyes closed which is known to increase alpha power, which may influence the generalisability of the alpha maps to an eyes open condition. However, given that our primary use of alpha was to act as a control, we believe that this should not affect the interpretability of the key findings of our study. Further information on the topography of theta channels is shown in *Figure 1—figure supplement 1*.

To investigate the regional distributions of frequency bands, we counted the occurrence of each dominant rhythm in gyri and lobes (see *Figure 1* and *Table 1*) by projecting the coordinates in MNI

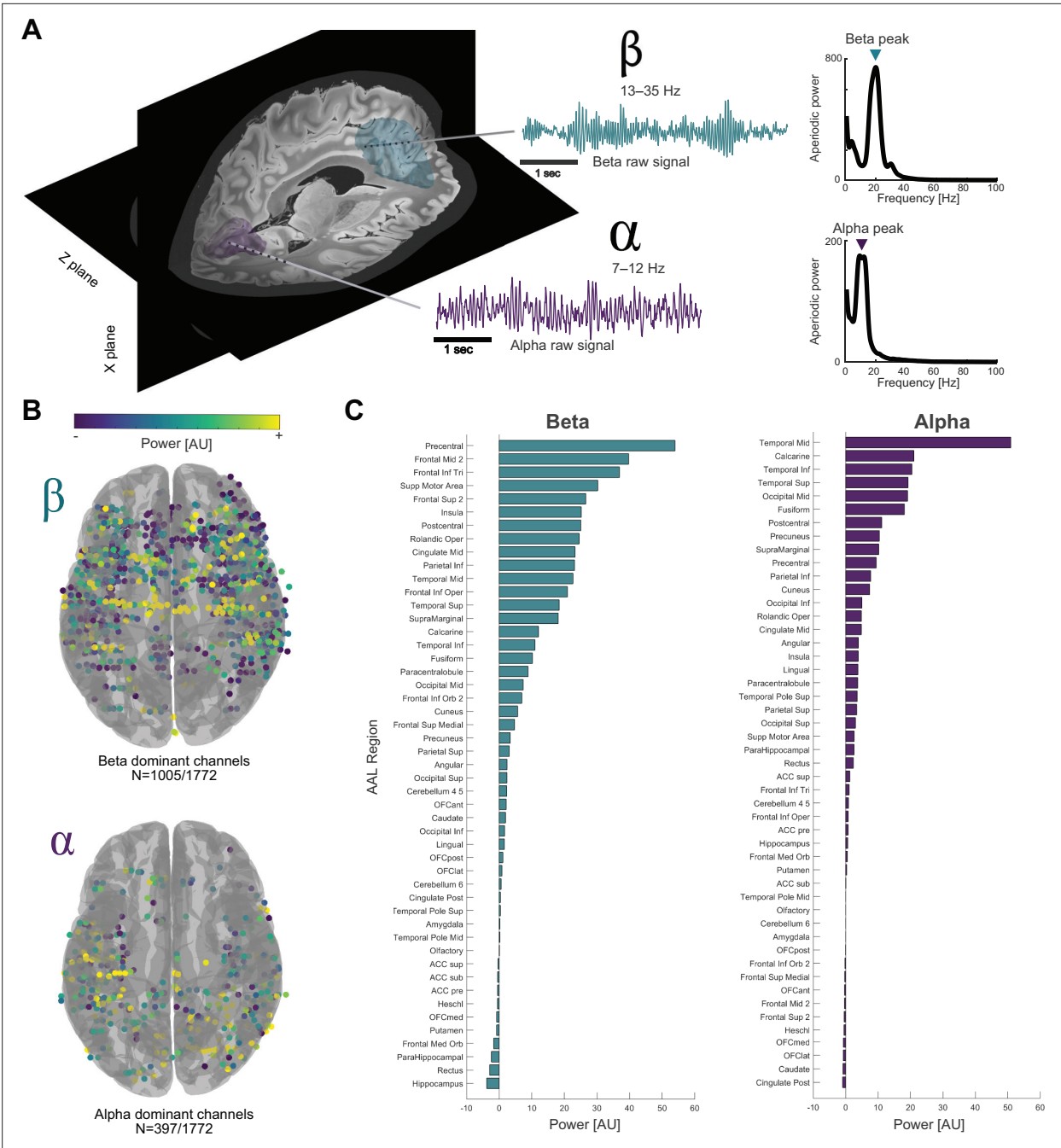

**Figure 1.** Beta activity is a widely distributed resting brain rhythm in invasive cortical recordings. (**A**) Exemplar stereoelectroencephalography (sEEG) electrodes with dominant beta (teal; 20 Hz) or dominant alpha (purple; 9 Hz) rhythms, alongside the respective raw signal and parametrised power spectrum. (**B**) 1005/1772 channels showed higher maximum peak power in beta (13–30 Hz) than theta (4–8 Hz), alpha (8–12 Hz), and gamma (30–100 Hz). 397/1772 were alpha-dominant (theta and gamma not shown). (**C**) Beta-dominant channels were distributed across the entire cortex, from occipital to frontal regions, with highest density in sensorimotor areas like precentral gyrus, frontal middle gyrus, frontal inferior gyrus, and supplementary motor area. Alpha channels were largely concentrated posteriorly in temporoparietal and occipital areas based on the automatic anatomical labelling atlas (*Rolls et al., 2020*). Neither alpha nor beta activity showed systematic hemispheric differences (p>0.05).

The online version of this article includes the following figure supplement(s) for figure 1:

**Figure supplement 1.** Cortical spread and amplitudes of theta peaks.

**Table 1.** Spread of invasive electrophysiological channels.

Given the low number of channels exhibiting resting-state gamma activity, we excluded those from this table.

| Lobe | Channels/lobe | ECoG | sEEG | Sum theta channels | | Sum alpha channels | | Sum beta channels | |
|---|---|---|---|---|---|---|---|---|---|
| Frontal | 616 | 122 | 494 | 73 | 11.80% | 38 | 6.20% | 458 | 73.40% |
| Temporal | 519 | 68 | 451 | 106 | 20.40% | 180 | 35.00% | 216 | 41.60% |
| Cingulate | 116 | 0 | 116 | 12 | 10.30% | 21 | 18.10% | 71 | 61.20% |
| Insula | 128 | 2 | 126 | 19 | 14.80% | 21 | 16.40% | 86 | 67.20% |
| Amygdala | 3 | 0 | 3 | 1 | 33.30% | 0 | 0% | 1 | 33.30% |
| Parietal | 272 | 52 | 220 | 50 | 18.40% | 73 | 26.80% | 139 | 51.10% |
| Occipital | 107 | 13 | 94 | 15 | 14.00% | 62 | 57.90% | 27 | 25.20% |
| Subcortex | 11 | 1 | 10 | 1 | 9.10% | 2 | 18.20% | 7 | 63.60% |

ECoG, electrocorticography; sEEG, stereoelectroencephalography.

The online version of this article includes the following source data for table 1:

**Source data 1.** sEEG channels (high and low beta calculated separately to calculate total beta peaks).

**Source data 2.** ECoG channels.

(Montreal Neurological Institute) space to the automatic anatomical labelling (AAL) atlas parcellation (*Rolls et al., 2020*). Notably, many channels showed more than one oscillatory peak, and all covered lobes had channels with dominant rhythms in each, theta, alpha, and beta frequency bands (see *Table 1*). Beta activity was the most frequent and most widely distributed resting rhythm across all lobes of the human brain, including frontal, temporal, cingulate, insula, and parietal lobes. Interestingly, the occipital lobe was the only brain region that showed higher percentage of alpha-dominant channels (53/94; 56.4%). The coverage of subdural ECoG grids or tissue penetrating sEEG electrodes could vary, but within overlapping regions, no major differences in the distribution of brain rhythms were observed for electrode type (see *Table 1—source data 1*, *Table 1—source data 2*).

## MRI connectomics reveal shared cortico-subcortical beta networks

Oscillatory synchronisation across brain regions may shape neural communication through modulation of synaptic sum potentials and neural firing (*Fries, 2015*). Oscillatory connectivity can be computed through correlation and modelling of phase and amplitude relationships between brain signals from multielectrode recordings. The spatial coverage of such computations, however, is inherently limited to brain regions that are accessible with the respective recording technique. For non-invasive M/EEG, the analysis of connectivity is limited to superficial cortical brain areas and can be biased by noise and volume conduction (*Pellegrini et al., 2023*). Invasive recordings in both human and animal studies provide more robust estimates of oscillatory connectivity but are limited to a selected number of channels from defined brain regions (*Engel et al., 2005*). Here, we aimed to investigate the whole-brain circuit representation of beta activity, which is impossible with non-invasive neurophysiology approaches. Instead, we tested the hypothesis that cortical brain areas that exhibit resting beta share a global brain network by means of MRI connectomics. For this, we derived the methodology from lesion and deep brain stimulation (DBS) network mapping (*Fox, 2018*), which has provided groundbreaking insights into the circuit nature of brain lesion and neurostimulation effects and applied it to reveal the shared network fingerprints of brain regions that exhibit a neurophysiological phenomenon; namely, dominant beta activity. For this, we followed a multimodal approach, investigating neurophysiologically identified brain regions by means of MRI connectomics, generating functional and structural connectivity maps for each brain signal recording location, as previously described (*Lofredi et al., 2021*). In brief, the MNI coordinates of the invasive electrode locations were used as seeds to compute whole-brain connectivity maps based on MRI data from 1000 healthy subjects as available through the Brain Genomics Superstruct project, and MRI data from 32 subjects as available through the Human Connectome Project (HCP) as part of the Lead Mapper pipeline (*Neudorfer et al., 2023*). While the connectivity maps estimated with functional MRI (fMRI) are based on the covariance blood oxygenation level-dependent signals that provide an estimate of functional coactivation, the structural

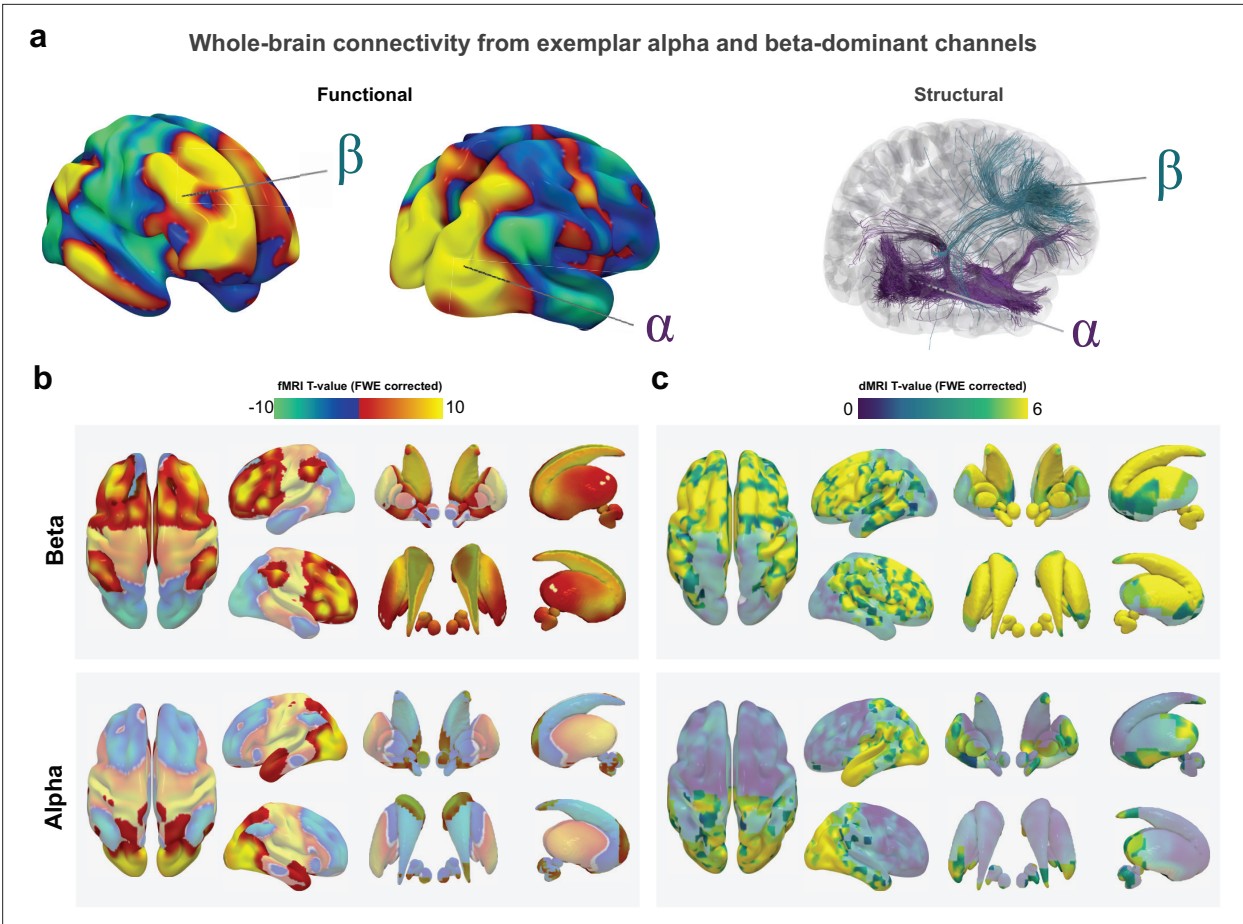

**Figure 2.** MRI connectomics reveal shared cortico-subcortical beta networks. Individual network fingerprints (**A**) seeded from representative beta (top) and alpha (bottom)-dominant electrodes highlight the methodological approach and showcase the distinct connectivity patterns that arise in dependence of the connected brain regions (for raw signal traces from these locations, see *Figure 1*). Aggregated functional (**B**) and structural (**C**) connectivity maps across all beta (top) and alpha (bottom)-dominant electrodes were subjected to mass univariate *t*-tests comparing beta vs. alpha networks with Statistical Parametric Mapping software and visualised as T-maps with significant clusters identified using family-wise error correction (shown opaque). Channel locations from beta-dominant channels were associated with more robust functional and structural connectivity to frontal cortex and the basal ganglia compared to alpha channel locations, which were more connected to the occipital cortex.

The online version of this article includes the following figure supplement(s) for figure 2:

**Figure supplement 1.** Unthresholded averages of functional and structural connectivity for alpha- and beta-dominant recording locations.

**Figure supplement 2.** Region-wise bar plots of functional connectivity values for every parcel based on the automatic anatomical labelling (AAL) atlas.

**Figure supplement 3.** Region-wise bar plots of structural connectivity values for every parcel based on the automatic anatomical labelling (AAL) atlas.

connectivity maps are based on diffusion-weighted MRI (dMRI) and provide a relative quantification of major anatomical white matter tracts. To allow for a statistical comparison of connectivity patterns, we compared voxel-wise connectivity estimates from recording locations with dominant beta rhythms to recording locations with dominant alpha rhythms using mass univariate two-sample *t*-tests in Statistical Parametric Mapping software, with a family-wise error (FWE)-corrected voxel threshold set to $p < 0.05$.

Our analysis revealed a shared whole-brain network for cortical brain regions that exhibit dominant beta oscillations, with marked and highly significant differences (*Figure 2*) to the control network for alpha oscillations, both in whole-brain functional and structural connectivity patterns (see *Figure 2—figure supplement 1* for unthresholded averages). Most prominently, the beta network showed significantly higher positive connectivity to prefrontal, cingulate, and parietal cortical areas and the ventral thalamus and basal ganglia, including substantia nigra, striatum, pallidum, and subthalamic nucleus. In contrast, whole-brain networks seeded from alpha-dominant channels showed a more widespread

distribution across the posterior cortical areas, occipital and temporal lobes in comparison to the beta networks. All connectivity values including their sign are shown in figures as brain region averages parcellated with the automatic anatomical labelling atlas in *Figure 2—figure supplements 2 and 3*.

## The connectomic beta network correlates with molecular markers of dopamine signalling

Invasive brain signal recordings in patients undergoing DBS for PD have robustly established a strong interaction between dopaminergic tone and amplitude of beta oscillations at rest, for example, by measuring changes in activity after withdrawal and administration of dopaminergic agents (*Lofredi et al., 2023*). However, this interaction has been largely neglected outside the field of PD, likely because research methods to establish direct relationships between beta and dopamine in healthy subjects are not readily available. With the present project, we aimed to characterise the circuit architecture of cortical beta oscillations. We found that cortical brain areas that exhibit resting beta activity share a common brain network. This network, unrelated to pathology, includes strong connectivity with the basal ganglia, the primary target for dopaminergic innervation from the substantia nigra and the ventral tegmental area in the human brain. To further quantify the potential relationship between the beta network and dopamine signalling, we made use of another openly available human neuroscience resource: a database of PET imaging, a molecular imaging technique that relies on intravenous injection of a radioactively labelled molecular tracer, which can be visualised to indicate brain areas with high uptake of neurotransmitters or related molecules in the brain (*Hansen et al., 2022*). Specifically, we analysed to what degree the spatial uptake patterns of dopamine, as measurable with fluorodopa (FDOPA; cohort average of 12 healthy subjects) and other dopamine signalling-related tracers that bind D1/D2 receptors (average of N = 17/44, respectively, healthy subjects), or the dopamine transporter (DAT; cohort average of N = 180 healthy subjects) were correlated with the unthresholded MRI connectivity maps. To reduce the dimensionality and complexity of this analysis, we first applied spatial normalisation and then averaged the abovementioned cohort images to yield a PET-based dopamine aggregate map (*Figure 3*). Next, we resliced the beta network map and the PET images to allow for a meaningful comparison using a combined parcellation with 476 brain regions that include cortex (*Huang et al., 2022*), basal ganglia (*He et al., 2020*), and cerebellum (*Diedrichsen et al., 2009*). Here, each parcel – which was a collection of voxels belonging to a particular brain region – from the connectivity map was correlated with the same parcel containing average neurotransmitter uptake from the respective PET scan (see *Figure 3A*). In this way, non-parametric Spearman's correlations between PET intensity and structural and functional connectivity maps of beta networks were obtained, which indicates to what degree the spatial distribution of connectivity is similar to the distribution of neurotransmitter uptake. This revealed a significant positive relationship between the beta functional and structural networks and dopamine uptake for cortex (rho = 0.22/0.40 for functional/structural networks, p=0.0001) and basal ganglia (rho = 0.50/0.33, p=0.0001), but not cerebellum (all p>0.05). These patterns were largely qualitatively reproduced when the analysis was performed on the individual tracers (see *Figure 3—figure supplement 1*). Additionally, we repeated the analysis with PET images of GABA uptake, to serve as an unspecific molecular control, which revealed no significant relationship between the beta functional network and GABA uptake cortically (rho = 0.04, p=0.22), in the basal ganglia (rho = 0.07, p=0.27) or cerebellum (rho = 0.10, p=0.22). The structural beta network revealed low correlation with GABA (n = 16 healthy individuals) uptake for cortical areas (rho = 0.17, p=0.0003), negative correlations for basal ganglia (rho = −0.55, p=0.0001), and no correlation for cerebellum (p>0.05). In summary, we found significant and specific overlap of the functional and structural connectivity maps seeded from beta-dominant cortical brain areas with the spatial patterns of dopamine uptake in the human brain.

## Discussion

The present study combined three openly available datasets of invasive neurophysiology, MRI connectomics, and molecular neuroimaging in human subjects to characterise the spatial distribution of brain regions exhibiting resting beta activity, their shared circuit architecture, and its correlation with molecular markers of dopamine signalling in the human brain. Our results can be broken down into three key insights: first, we demonstrate that the beta rhythm is the most widely distributed dominant

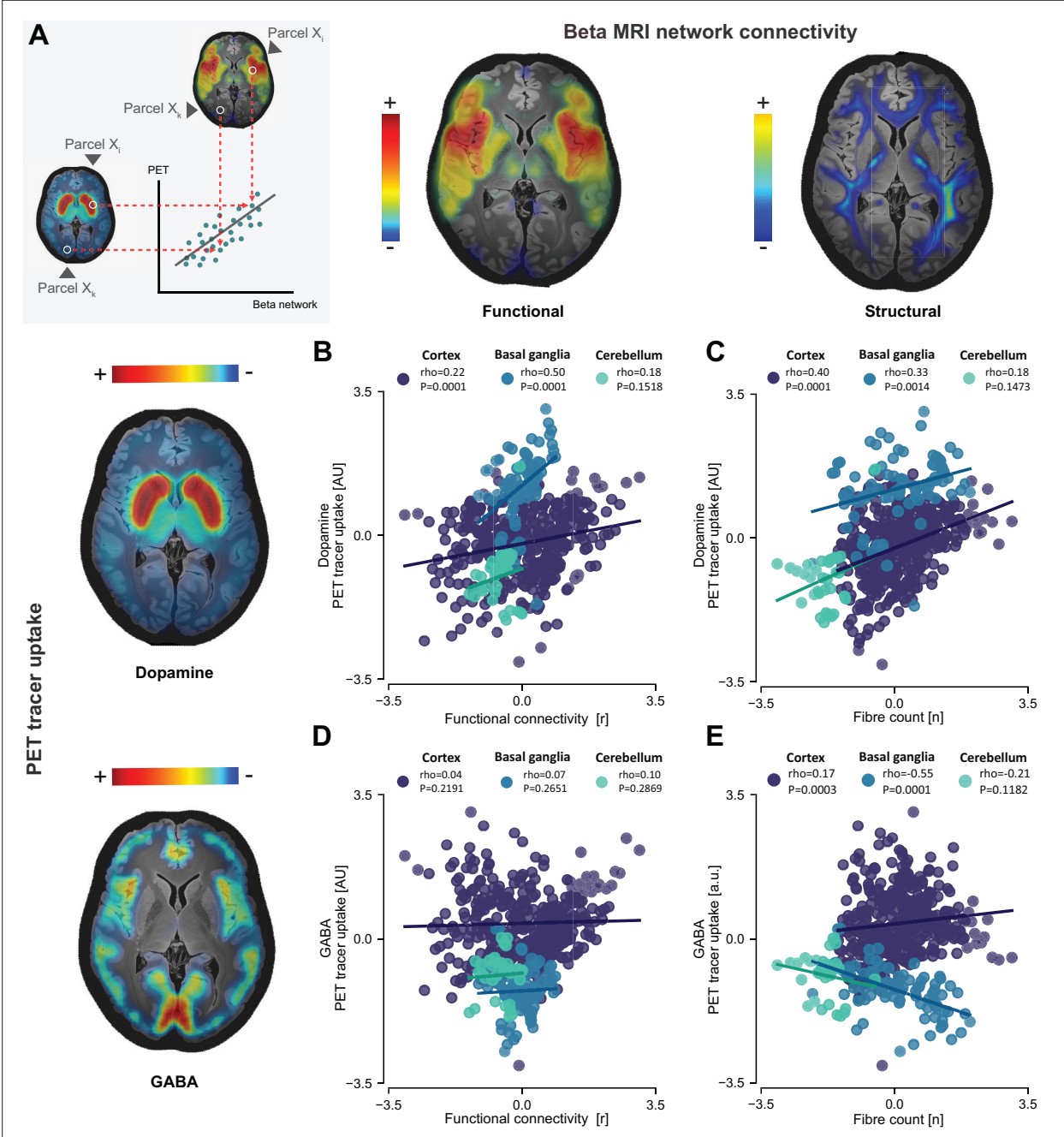

**Figure 3.** The connectomic beta network correlates with molecular markers of dopamine signalling. Dopamine uptake significantly correlates with beta functional connectivity. (**A**) A method schematic explaining how correlations were calculated. Each point on the scatterplot represents a parcellation from the compound brain atlas. For every parcel $X_i$ a correlation is calculated for that parcel in the PET scan and in the beta network (**B**) in cortex (rho = 0.22, p=0.0001) and basal ganglia (rho = 0.5, p=0.0001) but not cerebellum (p>0.05) and beta structural connectivity (**C**) in cortex (rho = 0.4, p=0.0001) and basal ganglia (rho = 0.33, p=0.001) but not cerebellum (p>0.05). GABA was used as a control molecule which revealed no significant correlations with beta functional connectivity (**D**) in cortex, basal ganglia, or cerebellum (p>0.05). Beta structural connectivity (**E**) revealed weaker positive correlation with GABA in cortex (rho = 0.17, p=0.0003), a negative correlation in the basal ganglia (rho = −0.55, p=0.0001), and no correlation in cerebellum (p>0.05).

The online version of this article includes the following figure supplement(s) for figure 3:

**Figure supplement 1.** Correlation plots for individual dopamine receptors.

neural synchronisation phenomenon, extending far beyond motor areas. Next, we show that these distributed areas share a common functional and structural whole-brain network that connects frontal, cingulate, and other large-scale brain areas with the basal ganglia. Finally, we show that this network shares significant spatial overlap with molecular markers of dopamine signalling. We interpret these findings as supportive evidence for the hypothesis that beta activity is generated and sustained by a cortico-subcortical brain network that is modulated by dopamine. This conceptualisation moves away from functionally specific domains towards a more holistic circuit understanding of beta activity as a brain circuit synchronisation phenomenon.

## Brain functions associated with beta oscillations share brain networks modulated by dopamine

Our study investigates the connectivity profile of cortical brain regions that spontaneously exhibit beta oscillations as their dominant brain rhythm. Our results suggest that one common denominator of brain regions that generate beta activity is their affiliation with a large-scale global brain network that is modulated by dopamine. However, we do not see this concept as contradictory to previous reports of functional correlates of beta activity in the domains of motor control, cognition, memory, and emotion (*Paulo et al., 2023*; *Hanslmayr et al., 2016*; *Alagapan et al., 2023*; *Neumann et al., 2023b*). Instead, we believe that the overlap in oscillatory activity across these domains can be conceptualised through a circuit definition of beta activity and its relationship to dopamine signalling. Foremost, in the motor domain there is direct evidence that pathological changes in dopamine causally modulates beta activity, as seen in human patients with PD (*Lofredi et al., 2023*), but also in animals lesioned with neurotoxins targeting dopaminergic neurons, such as 1-methyl-4-phenyl-1,2,3,6-tetrah ydropyridine (MPTP) in non-human primates (*Deffains and Bergman, 2019*) or 6-hydroxydopamine (6-OHDA) in rodents (*Sharott et al., 2005*). Moreover, the robust beta modulation that is elicited by voluntary action in sensorimotor cortex is long known. In our study, motor and sensorimotor cortex showed higher beta peak amplitudes than any other brain regions. Thus, it is tempting to infer that beta activity and its relation to dopamine are characteristic to the motor domain. On the other hand, numerous studies have reported specific changes in beta activity in relation to many other neuroscientific domains that define human ability. For example, it was shown that beta is implicated in working memory (*Miller et al., 2018*), utilisation of salient sensory cues (*Leventhal et al., 2012*), language processing (*Meyer, 2018*), motivation (*Pierrieau et al., 2023*), sleep (*Yin et al., 2023*), emotion recognition (*Jabbi et al., 2015*), mood (*Kirkby et al., 2018*), and may even serve as a biomarker for depressive symptom severity in the anterior cingulate cortex (*Clark et al., 2016*). Our connectomic characterisation of the beta network provides a network basis for these functions and highlights that similar synchronisation phenomena occur in brain areas associated with vastly different functions, such as prefrontal, cingulate, and parietal areas, which give rise to parallel striatal and other subcortical projections. In line with this more complex picture, direct measurement of dopamine concentration in non-human primates revealed specific interactions between dopamine release, beta oscillations, reward value, and motor control, depending on contextual information and striatal domain (*Schwerdt et al., 2020*). This shows that the relationship of dopamine and beta activity is not solely associated with either reward or movement and depends on where in the striatum beta activity is recorded. Thus, an obvious differentiator of these reported functions can be the specific brain region in which beta activity has been recorded from. However, the aim of our study was the opposite, namely to provide evidence for a shared network that unifies these functions.

An authoritative perspective article has proposed that the unifying function of beta oscillations is to signal the maintenance of 'status quo', for example, to indicate whether or not a motor or cognitive state should be changed or not (*Engel and Fries, 2010*). We hypothesise that this 'status quo' hypothesis could be equally or maybe even more adequately posed on the neural level. Namely, it could provide insights to what degree a certain activity pattern or synaptic connection is to be strengthened or weakened in light of neural learning. We propose that this putative function can be contextualised in a recently described framework of neural reinforcement that serves to orchestrate the re-entrance and refinement of neural population dynamics for the production of neural trajectories (*Athalye et al., 2020*). On the basis of our observed network architecture described for beta oscillations in this article, we speculate that the 'status quo' hypothesis and each of the abovementioned behavioural correlates can be summarised to reflect states with either loss or invigoration of neural reinforcement (*Cavallo*

*and Wolf-Julian, 2023*), the brain's intrinsic capacity to modulate the stability of population dynamics via cortico-striatal projections that are orchestrated by dopamine. To simplify this, maintenance of a behavioural, cognitive, or emotional 'status quo' may be reflected by beta activity on the neural circuit level, which in turn relates to the stability of neural dynamics in frontostriatal circuits that can be dynamically modulated by tonic and phasic changes of dopamine release. If beta is high, the neural dynamics remain stable. If beta is low, new neural trajectories may arise that lead to neural and behavioural state changes and neural learning. Thus, we speculate that the observed beta network and its relation to molecular markers of dopamine in this study may indicate that beta modulation occurs in situations that are associated with neural learning and optimisation of neural trajectories, or lack thereof. Notably, many functional correlates of beta activity could be summarised under the 'status quo' hypothesis, but some functions are not easily interpreted in this framework, such as the specific observations in the context of memory processing (*Miller et al., 2018*) or regulation of emotions and depression. On the other hand, the role for dopaminergic reinforcement and its implication in these processes is robustly established, both in memory (*Rossato et al., 2009*) and the pathophysiology of depression (*Dunlop and Nemeroff, 2007*). At this point, this discussion remains speculative, but advances in neuroscientific research methods that combine neurophysiology with neurotransmitter release may soon provide new evidence in favour or in opposition to these concepts.

## Implications for neurotechnology and clinical brain–computer interfaces

A key component of clinical brain–computer interfaces (BCIs) is the bidirectional communication between brain activity and closed-loop control of neurotherapies. The robust association of the hypodopaminergic state in PD with excessive beta synchrony has led to the development of a closed-loop neurostimulation therapy using beta activity as a feedback signal to adjust administration of DBS in real time to therapeutic demand (*Neumann et al., 2023a*). This approach, also termed adaptive DBS, was first demonstrated based on cortical beta activity that was used to adapt pallidal DBS in the MPTP non-human primate model of PD (*Rosin et al., 2011*). It was quickly translated to first experimental studies using subcortical beta signals in human patients (*Little et al., 2013*), followed by further research using more complex cortical and subcortical sensing setups and biomarker combinations (*Gilron et al., 2021*; *Oehrn et al., 2023*). Practically, beta activity is used in this novel therapeutic approach to infer the dopaminergic tone and consequential therapeutic demand in PD for rapid adaptation of therapeutic delivery to the individual state that the patients are facing in real time. Our study provides support for the validity of this approach by providing further evidence for a relationship of dopamine and beta activity at the global circuit level. More importantly, our findings may suggest that this relationship could be of greater utility of neurotechnology to treat other dopaminergic disorders. In fact, this notion is in part already backed by empirical data that shows that beta can be a valuable biomarker for depression severity as identified by machine learning algorithms in invasive neurophysiological recordings from brain implants (*Alagapan et al., 2023*). Given our observation that beta activity is the most widespread cortical brain rhythm, it is foreseeable that future experimental in-human trials aiming to test the utility of invasive neurophysiology for the adaptation of neurotherapies will be capable of tracking beta dynamics in higher frequency (sampling rate) and higher precision than any other brain rhythm. Our findings could encourage clinical neuroscience and neural engineering teams to conceptualise the dynamic fluctuations of beta oscillations in the context of cortico-striatal processing and dopaminergic modulation, which could lead to new hypothesis-driven treatment avenues. Specifically, our demonstration that the distributed occurrence of beta activity shares a brain network that is modulated by dopamine may motivate the use of beta oscillations to control BCIs or closed-loop neurostimulation for emerging indications related to dopaminergic dysfunction.

## Limitations

The most important limitation of this project is aggregated and derived nature of the used brain data. Three independent datasets from different study cohorts were leveraged to fuse the invasive neurophysiology with MRI and PET-based neuroimaging. On the one hand, the independence of the source data may be seen as a form of validation, for example, for functional and structural connectivity that showed highly overlapping findings while stemming from entirely different cohorts. On the other hand, precisely this aspect severely limits the generalisability of our results to the

level of the individual human subject. Similarly, the foundation of the results is built on patients undergoing brain surgery for epilepsy, a severe intractable brain disorder. But at the same time, the reported observations follow a long line of research and corroborate anecdotal or clinical knowledge that beta activity is a ubiquitous phenomenon. In this light, our study provides an unprecedented and methodologically robust quantitative approach by using only localised bipolar invasive recordings with oscillatory activity parametrised for optimal comparability. We believe that this rigorous neurophysiological methodology, alongside the fact that the brain signals have been meticulously controlled for pathological activity, by several experienced neurophysiologists, justifies its use and superiority over aggregates of non-invasive recordings from healthy subjects. Another limitation of our study is the fact that the statistical approach for the comparison of beta and alpha networks and even for multiple peaks within the same frequency band follows a winner-takes-all logic that is, by definition, a simplification as most areas will contribute to more than one spatio-spectrally distinct oscillatory network. Specifically, while multiple peaks within or across frequency bands could be present in each channel, we decided to allocate this channel to only the frequency band containing the highest peak amplitude. Alternatively, channels could have been allocated to multiple networks, contrasting channels that display vs. do not display peaks in a certain range. Alternative approaches could yield different results, for example, reusing channels for each peak that is observable and contrasting them to channels where such peak was not present. However, in our study the majority of channels exhibited beta activity even if peaks were of low amplitude, which we believe would have led to less interpretable results. Nevertheless, a previous study combining invasive basal ganglia neurophysiology with non-invasive cortical MEG recordings has identified the strongest oscillatory network of the internal pallidum to connect to frontal cortex in the beta frequency range, which is in line with our findings (*Neumann et al., 2015*). However, additional networks, such as an alpha network with the cerebellum and a theta network with the temporal lobe, were also observable in these parallel recordings. Thus, the presented MRI-based statistical maps in this study can be interpreted to reflect shared large-scale and robustly estimable whole-brain circuits of dominant beta vs. alpha oscillations but should not be misunderstood to reflect mutual exclusivity for the presence or communication in additional brain rhythms. In fact, when looking at individual parcels of structural connectivity, it is noticeable that both beta and alpha networks share strong connections to cingulate cortex. This is interesting because previous invasive neurophysiology studies in patients undergoing DBS surgery for depression have reported both excessive alpha and beta activity as potential biomarkers for depressive symptoms in the human cingulum (*Clark et al., 2016*), potentially related to differential modulation of neurotransmitters involved in the disease. One impactful study reported that beta oscillatory sub-networks of amygdala and hippocampus could reflect human variations in mood (*Kirkby et al., 2018*). This is interesting but highlights another relevant limitation of our study, namely that recordings in different areas were stemming from different patients, and thus, such subnetwork analyses on the oscillatory level could not be conducted.

Another limitation arises as a recurring controversy in fMRI researchis the issue insufficient knowledge about negative correlations and their exact scientific implications. At present, there are no straightforward answers about what such correlations actually mean, and therefore, we have been hesitant in drawing conclusions relating to the signs of fMRI connectivity, and instead, treat it as a spatial fingerprint.

It should also be noted that in this project the data are static, but beta oscillations and dopamine are dynamic (*Tinkhauser et al., 2017*). Therefore, while we cannot infer the direction of the relationship, it is very likely that the relationship would be negative, implying higher beta would mean lower dopamine. Finally, our study is limited to resting-state activity and thus does not allow for interpretation of functional changes with relation to behaviour or neural processes, which may individually be associated with other or additional brain rhythms. Most notably, gamma band oscillations are known to be reflective of local processing but notoriously absent in resting-state activity and thus are likely equally or even more ubiquitous than beta activity. However, we do not see this as a contradiction as resting beta activity provides insights into a different state of the cortical brain area, potentially more associated with global circuit synchrony, while gamma band activity may be more reflective of local activity in the cortical microcircuit. The aim of this study was to elucidate the circuit architecture of beta oscillations, which is why insights from this study for other frequency bands are limited. Future research investigating the specific circuits of theta, alpha, and gamma oscillations and their

relationship with neurotransmitter uptake could yield new important insights into the networks underlying human brain rhythms.

## Conclusion

In summary, our study suggests that the beta rhythm is the most widespread cortical oscillatory activity pattern in the resting brain that shares connectivity with the basal ganglia via a whole-brain network that is modulated by dopamine. We interpret our findings to provide a unifying circuit-based integration of previous reports on the functional role of beta oscillations. In the future, our findings may inspire and encourage the investigation of beta oscillations for the use as a feedback signal for neurotechnological interventions in the treatment of dopaminergic disorders.

## Methods

### Atlas of invasive electrophysiology

The collection of data, cohort, and electrophysiology of the MNI Open iEEG Atlas (https://mni-open-ieegatlas.research.mcgill.ca/) is as described in *Frauscher et al., 2018*. Intracranial EEG (iEEG) recordings from the healthy hemispheres of epilepsy patients were collected from three medical research centres. Then, 60 s EEG sections during resting wakefulness that were part of a controlled assessment across centres were chosen for analysis. Inclusion criteria for the recordings included that it must (1) contain at least one channel with normal activity, (2) have peri-implantation imaging, (3) be recorded after a minimum of 72 hr post-implantation for stereo-EEG electrodes or 1 week for subdural grids or strips, and (4) have a minimum sampling frequency of 200 Hz. The recordings were analysed to identify frequencies in the classical Berger frequency bands (0.3–70 Hz). Tissue containing channels exhibiting normal activity was classified as normal or healthy tissue. This meant it was outside of the zone of seizure onset, that is, no interictal epileptic discharge and no discharges or slow-wave anomalies were found there. Ethical approval was granted at the MNI as lead ethics organisation (REB vote: MUHC-15-950). Electrode locations were registered to a common stereotaxic space.

### Oscillatory analysis and neurophysiological preprocessing

Data from the iEEG atlas were preprocessed using the Statistical Parametric Mapping toolbox (https://www.fil.ion.ucl.ac.uk/spm12/). The original data contained zero padded intervals between signal fragments which were deleted. Next the data were Fourier transformed to obtain a time–frequency signal using Morlet wavelets with seven cycles. All time–frequency signals were averaged over time to obtain one frequency spectrum of 60 s sections of iEEG measurements. The resulting power spectra were quantified in terms of spectral peak properties and background-level activity using FOOOF (*Donoghue et al., 2020*) with the following parameters; limits for possible peak widths were [0.5, 12 Hz], the maximum number of detectable peaks were set to infinite so as to not limit the number of peaks detected, and the minimum peak height was 0 while the peak threshold was within 2 standard deviations of modelled power. Peak height was extracted using the pw parameter, which depicts peak amplitude after subtraction of any aperiodic activity. In case of multiple peaks within the same region, we used only the highest peak amplitude. Those extracted were z-scored within electrodes, within subjects, and within the whole dataset to quantify neural activity from several frequency bands of the spectrum. Frequency-specific activity was quantified using maximum peak amplitude for specific frequency ranges. The maximum peak amplitude was extracted from the z-scored parameters for four frequency bands; theta (4–8 Hz), alpha (8–12 Hz), beta (13–35 Hz), and gamma (30–100 Hz).

### Connectivity analysis

Resting-state functional connectivity data was collected from 1000 healthy individuals using a 3T Siemens MRI scanner as part of the Brain Genomics Superstruct Project. Preprocessing included global signal regression and spatial smoothing at 6 mm full width at half maximum. The group connectome for structural connectivity was obtained from multishell diffusion-weighted and T2-weighted imaging data subjects from the HCP (*Setsompop et al., 2013*). The imaging data were acquired using a custom MRI scanner with enhanced gradients. Whole-brain tractography fibre sets were generated using a generalised q-sampling imaging algorithm implemented in DSI Studio. Connectivity analyses were performed in MATLAB using normative connectomes calculated with Lead-Mapper (*Horn and*

*Kühn, 2015*). Three-dimensional spherical nifti files representing channel locations were extracted and used as seeds to calculate connectivity maps.

The fMRI and dMRI data underwent processing using the Lead-Connectome Mapper. Every contact location from the iEEG atlas was represented by a spherical seed nifti image ($r$ = 5 mm) To minimise the impact of noise on the results, all maps underwent smoothing with a Gaussian kernel (standard deviation = 8 mm). Subsequently, the fMRI maps were masked with a grey matter mask to avoid the inclusion of white matter correlations, which would have introduced artefacts.

A functional connectivity map, that is, a mapping of the functional connectivity (based on a normative connectome) between a seed region and every other brain region, was derived for each individual channel (N = 1772). To do so, each channel was used as a seed region and its respective functional connectivity map was calculated, resulting in 1772 connectivity maps, one for every channel. These maps were subdivided based on the channel's dominant frequency band. This resulted in 277 functional connectivity maps for the theta, 397 for the alpha, and 1005 for the beta band. Group-level statistics were performed for connectivity maps of each frequency band. All functional connectivity maps were averaged in SPM ImCalc to reveal a group-level functional network for each frequency band. To investigate differences between beta-dominant and alpha-dominant functional connectivity networks, a two-sample $t$-test was calculated for the condition where beta was greater than alpha and vice versa using SPM. Here, the connectivity maps from each dominant channel (1005 beta functional connectivity maps and 397 alpha connectivity maps). Estimation of model parameters yielded $t$-values for each voxel, indicating the strength and direction of differences between the two contrasts (beta > alpha, alpha > beta). To address the issue of multiple comparisons, we applied FWE correction, adjusting significance thresholds such that only voxels with $p<0.05$ would be included.

Like functional connectivity maps, structural connectivity maps (N = 1772) were calculated for each individual channel. These were again subdivided based on dominant frequency band per channel and gave a resulting set of structural connectivity maps for each frequency band of 277 (theta), 397 (alpha), and 1005 (beta). Each map contained a set of fibre tracts that were identified as correlated with the specific channel through the structural connectivity analysis. Therefore, to investigate group-level statistics, the maps had to be summed and not averaged as in the case of functional connectivity. The resulting structural networks for each frequency band were hence a sum calculated on SPM ImCalc of all identified fibres for all channels for a given frequency band.

## Neurotransmitter uptake

To investigate dopamine uptake, an open dataset of neurotransmitter uptake was used for an open data set of PET data measuring neurotransmitter uptake (*Hansen et al., 2022*). Exclusively healthy samples of subjects were selected for various dopamine receptors such as D1 (*Kaller et al., 2017*) (healthy controls [HC] = 13), D2 (*Alakurtti et al., 2015*; *Smith et al., 2019*) (HC = 34), the DAT (*Dukart et al., 2018*; *Sasaki et al., 2012*) (HC = 180), and FDOPA (*García Gómez et al., 2018*) (HC = 12). To study the broad effects of dopamine, all dopamine receptor PET scans were resliced and averaged using the SPM ImCalc function. A spatial normalisation was then applied to finally create a unified aggregate image of all available dopamine receptors. A custom master parcellation in MNI space was created in MATLAB using SPM functions by combining three existing parcellations to include cortical regions (*Huang et al., 2022*), structures of the basal ganglia (*He et al., 2020*), and cerebellar regions (*Diedrichsen et al., 2009*). Regions that were (partially) overlapping between the atlases were only selected once. The final compound parcellation had 476 regions in total. This parcellation was applied to both PET and unthresholded structural and functional connectivity maps using SPM and custom code. This allowed for the calculation of spatial correlations, providing a statistical measure of spatial similarity of the PET intensity and MRI connectivity distributions. For this, Spearman's ranked correlations were used to calculate correlations between the PET images, such as the dopamine aggregate map and both functional and structural beta connectivity networks (*Figure 3*). The analysis was repeated for individual tracers showing similar results (*Figure 3—figure supplement 1*). Finally, to validate these results, a control analysis was performed using a GABA PET scan from the same open dataset of neurotransmitter uptake following the same pipeline (*Figure 2A and B*).

## Acknowledgements

We thank Alessia Cavallo, Dr. Till Dembek, Dr. Konstantin Butenko, and Dr. Garance Meyer for their support in the preparation of the manuscript. The study was funded by the Hertie Network of Excellence in Clinical Neuroscience fellowship to WJN. WJN further received funding from the European Union (ERC, ReinforceBG, project 101077060), Deutsche Forschungsgemeinschaft (DFG, German Research Foundation) – Project-ID 424778381 – TRR 295, and the Bundesministerium für Bildung und Forschung (BMBF, project FKZ01GQ1802).

## Additional information

### Competing interests

Wolf Julian Neumann: received honoraria for consulting from InBrain - Neuroelectronics that is a neurotechnology company and honoraria for talks from Medtronic that is a manufacturer of deep brain stimulation devices unrelated to this manuscript. The other authors declare that no competing interests exist.

### Funding

| Funder | Grant reference number | Author |
| --- | --- | --- |
| European Research Council | ReinforceBG project 101077060 | Wolf Julian Neumann |
| Deutsche Forschungsgemeinschaft | Project-ID 424778381 - TRR 295 | Wolf Julian Neumann |
| Bundesministerium für Bildung und Forschung | FKZ01GQ1802 | Wolf Julian Neumann |

The funders had no role in study design, data collection and interpretation, or the decision to submit the work for publication.

### Author contributions

Meera Chikermane, Conceptualization, Data curation, Formal analysis, Visualization, Methodology, Writing – original draft, Project administration, Writing – review and editing; Liz Weerdmeester, Data curation; Nanditha Rajamani, Visualization, Methodology; Richard M Köhler, Timon Merk, Visualization; Jojo Vanhoecke, Methodology; Andreas Horn, Supervision, Methodology, Writing – review and editing; Wolf Julian Neumann, Conceptualization, Supervision, Funding acquisition, Investigation, Writing – original draft, Project administration, Writing – review and editing

### Author ORCIDs

Meera Chikermane https://orcid.org/0000-0003-1800-1346
Liz Weerdmeester https://orcid.org/0009-0005-3024-4086
Richard M Köhler https://orcid.org/0000-0002-5219-1289
Timon Merk https://orcid.org/0000-0003-3011-2612
Jojo Vanhoecke https://orcid.org/0000-0002-9857-1519
Wolf Julian Neumann https://orcid.org/0000-0002-6758-9708

Reviewer #1 (Public review): https://doi.org/10.7554/eLife.97184.3.sa1
Reviewer #2 (Public review): https://doi.org/10.7554/eLife.97184.3.sa2
Reviewer #3 (Public review): https://doi.org/10.7554/eLife.97184.3.sa3
Author response https://doi.org/10.7554/eLife.97184.3.sa4

## Additional files

### Supplementary files
• MDAR checklist

## Data availability

All data used in the course of this work were publicly available open datasets and can be found online. The iEEG atlas is available on the McGill database and can be accessed via their publication (https://www.mcgill.ca/frauscher-lab/files/frauscher-lab/frauscher_brain2018.pdf). The normative connectomes are available within the Lead-DBS software (https://www.lead-dbs.org/). The PET scans for dopamine uptake as well as their data pipeline are available on the Network Neuroscience Lab GitHub repository (https://github.com/netneurolab/hansen_receptors; *Hansen, 2022*). All original code used to preprocess and analyze these data, and create the main and supplementary results is openly available on a GitHub repository (https://github.com/meera-cm/iEEG_connectomics, copy archived at *Chikermane, 2024*). Additional code from the open datasets is available at (https://github.com/leaddbs/leaddbs; *Horn et al., 2019*; *Li, 2024*) (Lead-DBS and Lead-Connectome Mapper) and https://github.com/netneurolab/hansen_receptors/tree/main/code (Hansen receptors).

The following dataset was generated:

| Author(s) | Year | Dataset title | Dataset URL | Database and Identifier |
|---|---|---|---|---|
| Cohen A, Al-Fatly B, Horn A | 2023 | GSP1000 Preprocessed Connectome for Lead DBS (V2) | https://doi.org/10.7910/DVN/KKTJQC | Harvard Dataverse, 10.7910/DVN/KKTJQC |

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
