## [Editor Report · eLife Assessment]

This study poses an **important** step forward in understanding the brain-network embedding of beta oscillations. The study advances our circuit-level understanding of the pathophysiology associated with dopaminergic alterations in psychiatric or neurological disorders. The study provides **compelling** evidence that beta oscillations across the neocortex and basal ganglia map onto shared functional and structural networks that show significant positive correlations with dopamine receptors.

---

## [Referee Report · Reviewer #1 (Public review)]

The study by Chikermane and colleagues investigates functional, structural, and dopaminergic network substrate of cortical beta oscillations (13-30 Hz). The major strength of the work lies in the methodology taken by the authors, namely a multimodal lesion network mapping. First, using invasive electrophysiological recordings from healthy cortical territories of epileptic patients they identify regions with highest beta power. Next, they leverage open access MRI data and PET atlases and use the identified high-beta regions as seeds to find (1) the whole-brain functional and structural maps of regions that form the putative underlying network of high-beta regions and (2) the spatial distribution of dopaminergic receptors that show correlation with nodal connectivity of the identified networks. These steps are achieved by generating aggregate functional, structural, and dopaminergic network maps using lead-DBS toolbox, and by contrasting the results with those obtained from high-alpha regions. The main findings are:

(1) Beta power is strongest across frontal, cingulate, and insular regions in invasive electrophysiological data, and these regions map onto a shared functional and structural network.

(2) The shared functional and structural networks show significant positive correlations with dopamine receptors across cortex and basal ganglia (which is not the case for alpha, where correlations are found with GABA).

---

## [Referee Report · Reviewer #2 (Public review)]

Summary:

This is a very interesting paper that leveraged several publicly available datasets: invasive cortical recording in epilepsy patients, functional and structural connectomic data, and PET data related to dopaminergic and gaba-ergic synapses. These were combined to create a unified hypothesis of beta band oscillatory activity in the human brain. They show that beta frequency activity is ubiquitous, and does not just occur in sensorimotor areas. Cortical regions where beta oscillations predominated had high connectivity to regions that are high in dopamine re-update.

Strengths:

The authors leverage and integrate three publicly available human brain datasets in a creative way. These public datasets are powerful tools for human neuroscience, and it is innovative to combine these three types of data into a common brain space to generate novel findings and hypotheses. Findings are nicely controlled by separately examining cortical regions where alpha predominates (which have a different connectivity pattern). GABA uptake from PET studies is used as a control for the specificity of the relationship between beta activity and dopamine uptake. There is much interest in synchronized oscillatory activity as a mechanism of brain function and dysfunction, but the field is short on unifying hypotheses of why particular rhythms predominate in particular regions. This paper contributes nicely to that gap. It is ambitious in generating hypotheses, particularly that modulation of beta activity may be used as a "proxy" for modulating phasic dopamine release.

Weaknesses:

As the authors point out, the use of normative data is excellent for exploring hypotheses but does not address or explore individual variations which could lead to other insights. It is also biased to resting state activity; maps of task related activity (if they were available) might show different findings.

Challenges:

In the Discussion, the authors do a fairly deep dive into the implications of their findings, particularly with respect to the hypothesis that beta band activity "preserves the status quo", and with respect to the use of beta band activity in controlling brain-machine interfaces. Mechanistically and theoretically oriented readers might gain rewarding new insights by a careful read of the discussion, but full appreciation of their deep dive may require real time interaction with the authors.

---

## [Referee Report · Reviewer #3 (Public review)]

Summary:

In this paper, Chikermane et al. leverage a large open dataset of intracranial recordings (sEEG or ECoG) to analyze resting state (eyes closed) oscillatory activity from a variety of human brain areas. The authors identify a dominant proportion of channels in which beta band activity (12-30Hz) is most prominent, and subsequently seek to relate this to anatomical connectivity data by using the sEEG/ECoG electrodes as seeds in a large set of MRI data from the human connectome project. This reveals separate regions and white matter tracts for alpha (primarily occipital) and beta (prefrontal cortex and basal ganglia) oscillations. Finally, using a third available dataset of PET imaging, the authors relate the parcellated signals to dopamine signaling as estimated by spatial uptake patterns of dopamine, and reveal a significant correlation between the functional connectivity maps and the dopamine reuptake maps, suggesting a functional relationship between the two.

Strengths:

Overall, I found the paper well justified, focused on an important topic and interesting. The authors' use of 3 different open datasets was creative and informative, and it significantly adds to our understanding of different oscillatory networks in the human brain, and their more elusive relation with neuromodulator signaling networks by adding to our knowledge of the association between beta oscillations and dopamine signaling. Even my main comments about the lack of a theta network analysis and discussion points are relatively minor, and I believe this paper is valuable and informative.

Weaknesses:

The analyses were adequate, and the authors cleverly leverage these different datasets to build an interesting story. The main aspect I found missing (in addition to some discussion items, see below) was an examination of the theta network. Theta oscillations have been involved in a number of cognitive processes including spatial navigation and memory, and have been proposed to have different potential originating brain regions, and it would be informative to see how their anatomical networks (e.g. as in Fig. 2) look like under the author's analyses.

The authors devote a significant portion of the discussion to relating their findings to a popular hypothesis for the function of beta oscillations, the maintenance of the "status quo", mostly in the context of motor control. As the authors acknowledge, given the static nature of the data and lack of behavior, this interpretation remains largely speculative and I found it a bit too far-reaching given the data shown in the paper. In contrast, I missed a more detailed discussion on the growing literature indicating a role for beta in mood (e.g. in Kirkby et al. 2018), especially given the apparent lack of hippocampal and amygdala involvement in the paper, which was surprising.

---

## [Author Response]

The following is the authors’ response to the original reviews.

**Reviewer #1 (Public Review):**
The study by Chikermane and colleagues investigates the functional, structural, and dopaminergic network substrates of cortical beta oscillations (13-30 Hz). The major strength of the work lies in the methodology taken by the authors, namely a multimodal lesion network mapping. First, using invasive electrophysiological recordings from healthy cortical territories of epileptic patients they identify regions with the highest beta power. Next, they leverage open-access MRI data and PET atlases and use the identified high-beta regions as seeds to find (1) the whole-brain functional and structural maps of regions that form the putative underlying network of high-beta regions and (2) the spatial distribution of dopaminergic receptors that show correlation with nodal connectivity of the identified networks. These steps are achieved by generating aggregate functional, structural, and dopaminergic network maps using lead-DBS toolbox, and by contrasting the results with those obtained from high-alpha regions.The main findings are:(1) Beta power is strongest across frontal, cingulate, and insular regions in invasive electrophysiological data, and these regions map onto a shared functional and structural network. (2) The shared functional and structural networks show significant positive correlations with dopamine receptors across the cortex and basal ganglia (which is not the case for alpha, where correlations are found with GABA).Nevertheless, a few clarifications regarding the choice of high-power electrodes and distributions of functional connectivity maps (i.e., strength and sign across cortex and sub-cortex) can help with understanding the results.

We thank the reviewer for this critical expert assessment.

**Reviewer #1 (Recommendations For The Authors):**
To potentially enhance the quality of the manuscript in the current version, I kindly ask the authors to address the following points:Major:(A) Power analysis of electrophysiological data(1) How were significant peaks identified exactly? I understand that the authors used FOOOF methodology to estimate periodic components of brain activity.

Thank you for pointing us to this lack of clarity. The application of FOOOF consists of the fitting of a one-over-f curve that delineates the aperiodic component followed by the definition of gaussians to fit periodic activity. This allows for extraction of periodic peak power estimates that are corrected for offset and exponent of the one-over-f or non-oscillatory aperiodic component in the spectrum (further information can be found here). We included all peaks that could be fitted using the process.

How about aperiodic components (Figure 1, PSD plots)?

We share the interest in aperiodic activity with the reviewer. However, given that the primary aim of this study was the description of beta oscillations and the methodology and results presentation is already very complex, we did not include the analysis of aperiodic activity in this manuscript. This could be done in the future and it would surely be interesting to visualize the whole brain connectomic fingerprints of aperiodic exponent and offset. With regard to the purely anatomical description of nonoscillatory aperiodic activity we would like to refer to Figure 8 in Frauscher et al. Brain 2018 (https://doi.org/10.1093/brain/awy035) where this is described. We have decided not to include additional information on this matter, because (a) we felt that this would further convolute the results and discussion without directly addressing any of the hypotheses and aims that we set out to tackle and (b) the interpretation of aperiodic activity is still a matter of intense research with conflicting results, which warrants very careful considerations of many aspects that again would go beyond the scope of this paper.

In addition, to what degree would the results change if one identified the peaks relative to sites with no peak, similar to Frauscher et al.

Beta activity, the oscillation of interest in our analysis is ubiquitous in the brain. In fact, of 1772 channels, only 21 channels did not exhibit a beta peak detectable with FOOOF. Thus, a comparison of 1751 against 21 would not yield meaningful results. We have therefore decided to focus on the channels in which beta activity is the strongest and dominant observable oscillation.

If the FOOOF approach has some advantages, these should be pointed out or discussed.

FOOOF indeed has the advantage that it provides an objective and reproducible estimation of peak oscillatory activity that accounts for differences in aperiodic activity. To the best of our knowledge, there is no other approach that is nearly as well documented, validated and computationally reproducible.

Changes in manuscript: We have now further clarified the definition of peak amplitudes in the results and methods section and have discussed the use of alternative measures in the limitations section of our manuscript.

Results: “The frequency band with the highest peak amplitude was identified using the extracted peak parameter (pw) for each channel and depicted as the dominant rhythm for the respective localisation (Figure 1).”

Methods: “Peak height was extracted using the pw parameter, which depicts peak amplitude after subtraction of any aperiodic activity.”

Discussion: “Alternative approaches could yield different results, e.g. reusing channels for each peak that is observable and contrasting them to channels where such peak was not present. However, in our study the majority of channels exhibited beta activity, even if peaks were of low amplitude, which we believe would have led to less interpretable results.”

(2) How exactly do the authors deal with channels with more than one peak? Some elaboration on this and how this could potentially impact the results would be appreciated. Sorry if I have missed it.

Indeed, a description of this was lacking so we are very thankful that the reviewer pointed this out. The maximum peak amplitude method was a winner-takes-all approach where in the case of multiple peaks, the peak with the higher amplitude was chosen. This method of course has drawbacks in the form of lost or disregarded peaks and remains a limitation to this study.

Changes in manuscript: We have now clarified this in the methods and results sections, which now read:

Methods: “In case of multiple peaks within the same region, we used only the highest peak amplitude.”

Results: “In case of multiple peaks within the same frequency band, we focused the analysis on the peak with the highest amplitude.”

And added the following to the Limitations section of the discussion:

“Another limitation in our study is the fact that the statistical approach for the comparison of beta and alpha networks and even for multiple peaks within the same frequency band follows a winner takes all logic that is, by definition, a simplification, as most areas will contribute to more than one spatiospectrally distinct oscillatory network. Specifically, while multiple peaks within or across frequency bands could be present in each channel, we decided to allocate this channel to only the frequency band containing the highest peak amplitude.”

(B) Network mapping(1) Knowing that fMRI data are preprocessed by regressing the global signal, there are negative correlations across the functional networks. Unfortunately, the distribution, sign, and strength of the correlations are not quantitatively shown in any of the plots. Thus, it is unclear whether, e.g., corticocortical vs. subcortico-cortical correlations differ in strength and/or sign. I think this additional information is important for better understanding the up/down-regulation of beta, e.g., by DA signaling. Some discussion around this point in addition would be insightful, I think.

The referee is touching upon a very important and difficult point, which we have considered very carefully. Global signal regression is a controversial topic and the neurophysiological basis of negative correlations remains to be elucidated. We can justify our use of this approach based on an expert consensus described in Murphy & Fox 2017 (https://doi.org/10.1016%2Fj.neuroimage.2016.11.052), which highlights that global signal regression can improve the specificity of positive correlations, improve the correspondence to anatomical connectivity. The truth however is that, we relied on it, because it is the more commonly used and validated approach used in lesion network and DBS connectivity mapping and implemented in the Lead Mapper pipeline. Indeed all connectivity estimates are shown in Supplementary figure 3. We remain hesitant to raise the focus to these points, because of the uncertain underlying neural correlates. However, when looking at the values, it is interesting to note that most key regions of interest exhibit positive connectivity values.

Changes in manuscript: We now point to the supplement containing all connectivity values in the results section more prominently: “All connectivity values including their sign are shown in figures as brain region averages parcellated with the automatic anatomical labelling atlas in supplementary figures 2&3.”

(2) I assume no thresholding is applied to the functional connectivity maps (in a graph-theoretical sense). Please clarify this is also related to the comment above, in particular, the strength of correlations.

Indeed, we demonstrate SPM maps using family wise error corrected stats in figure 2, but all further analyses were performed on unthresholded maps as correctly pointed out by the referee.

Changes in manuscript:

Results: “Specifically, we analysed to what degree the spatial uptake patterns of dopamine, as measurable with fluorodopa (FDOPA; cohort average of 12 healthy subjects) and other dopamine signalling related tracers that bind D1/D2 receptors (average of N=17/44 respectively healthy subjects) or the dopamine transporter (DAT; cohort average of N=180 healthy subjects) were correlated with the unthresholded MRI connectivity maps.”

Methods: “This parcellation was applied to both PET and unthresholded structural and functional connectivity maps using SPM and custom code.”

Minor(1) Methods, Connectivity analysis: The description of (mass-univariate) GLM analysis is confusing. The maps underwent preprocessing? Which preprocessing steps are meant here? What is the dependent variable and what are the predictors exactly?

We thank the reviewer for catching this error in our methods. We apologise for the confusion and mistake and thank the reviewer for catching it. Indeed, we have used t-tests without further preprocessing instead of a GLM.

Changes in manuscript: The respective section has been removed from the methods section and intermediate steps have been clarified. The section now reads: “To investigate differences between beta dominant and alpha dominant functional connectivity networks, a two sample t-test was calculated for the condition where beta was greater than alpha and vice versa using SPM. Here, the connectivity maps from each dominant channel (1005 beta functional connectivity maps and 397 alpha connectivity maps) Estimation of model parameters yielded t-values for each voxel, indicating the strength and direction of differences between the two contrasts (beta > alpha, alpha > beta). To address the issue of multiple comparisons, we applied Family-Wise Error (FWE) correction, adjusting significance thresholds such that only voxels with p < 0.05 would be included.”

(2) I encourage the authors to find a better (visual) way of reporting Table 1, to make the main observations easier to grasp and compare (maybe a two-dimensional bar plot? Or color-coding the cells?)

Reply: Thank you for your suggestion to improve the table, the new table is adjusted to the recommended changes to make it more readable.

**Reviewer #2 (Public Review):**
Summary:This is a very interesting paper that leveraged several publicly available datasets: invasive cortical recording in epilepsy patients, functional and structural connectomic data, and PET data related to dopaminergic and gaba-ergic synapses. These were combined to create a unified hypothesis of beta band oscillatory activity in the human brain. They show that beta frequency activity is ubiquitous, not just in sensorimotor areas, and cortical regions where beta predominated had high connectivity to regions high in dopamine re-uptake.Strengths:The authors leverage and integrate three publicly available human brain datasets in a creative way. While these public datasets are powerful tools for human neuroscience, it is innovative to combine these three types of data into a common brain space to generate novel findings and hypotheses. Findings are nicely controlled by separately examining cortical regions where alpha predominates (which have a different connectivity pattern). GABA uptake from PET studies is used as a control for the specificity of the relationship between beta activity and dopamine uptake. There is much interest in synchronized oscillatory activity as a mechanism of brain function and dysfunction, but the field is short on unifying hypotheses of why particular rhythms predominate in particular regions. This paper contributes nicely to that gap. It is ambitious in generating hypotheses, particularly that modulation of beta activity may be used as a "proxy" for modulating phasic dopamine release.Weaknesses:As the authors point out, the use of normative data is excellent for exploring hypotheses but does not address or explore individual variations which could lead to other insights. It is also biased to resting state activity; maps of task-related activity (if they were available) might show different findings.The figures, results, introduction, and methods are admirably clear and succinct but the discussion could be both shorter and more convincing.
**Reviewer #2 (Recommendations For The Authors):**
The tone of the discussion is excessively lofty and abstract, and hard to follow in places. Specific examples in comments to authors below.

We thank the reviewer for their positive assessment and their constructive feedback on the discussion. Also in light of the other reviewers we have made a sincere effort to shorten, restructure and improve the discussion. Additionally, we have addressed all the specific comments the reviewer had below. We appended each change to the manuscript where appropriate below and have addressed all comments in the main text. Having that said, we see this paper and discussion to provide our most up-to-date and personal perspective on a correct concept on the interplay of beta oscillations and dopamine that is generalizable. Providing a concept that is so generalizable is very challenging and so far very few authors have even attempted this. One notable exception is the “status quo” concept by Fries & Engel. While we will do our very best to address the comments, we have decided not to deviate from our initial ambition to provide a discussion on a generalizable concept. Naturally such a concept must be very complex and therefore it will be hard to understand in parts. Through the revision, we hope that the readability and comprehensibility has improved, while it provides an in-depth perspective and hypothesis on how beta oscillations, dopamine and their brain circuits may facilitate brain function. Nevertheless, we want to express our honest gratitude for the thoroughness with which the reviewer has read and scrutinized our paper. The review clearly tells that the reviewer had the ambition to follow and understand what we were trying to convey, which can be rare nowadays. We are truly thankful for this.

The first sentence is not quite true, as invasive neurophysiology was not, and cannot be, done in healthy humans. "The present study combined three openly available datasets of invasive neurophysiology, MRI connectomics, and molecular neuroimaging in healthy humans to characterise the spatial distribution of brain regions exhibiting resting beta activity, their shared circuit architecture, and its correlation with molecular markers of dopamine signaling in the human brain."

Changes in manuscript: We have now removed the “healthy” from the respective sentence.

"Our results motivate to conceptualise the capacity to generate.... This is not clear.

Changes in manuscript: “Our results suggest that one common denominator of brain regions that generate beta activity, is their affiliation with beta oscillations as a feature that arises from a largescale global brain network that is modulated by dopamine.”

"Similarly, the robust beta modulation that is elicited by voluntary action in sensorimotor cortex and its correlation with motor symptoms of Parkinson's disease is long known" - the association between movement-related cortical beta desynchronization and Parkinson's motor signs is not well described - could the authors specify and reference this?

We thank the reviewer for pointing out this lack of clarity. We meant that independently beta is known for “movement” and for “movement disorders” and not “movement in movement disorders”. Having that said, there are some studies that suggest that beta ERD is altered in PD (e.g.https://doi.org/10.1093/cercor/bht121), but saying that this is “long known” would be an overstatement and was not our intention. We rephrased this sentence accordingly.

Changes in manuscript: The sentence now reads: “Moreover, the robust beta modulation that is elicited by voluntary action in sensorimotor cortex and its correlation with motor symptoms of Parkinson’s disease is long known.”

"...first fast-cyclic voltammetry experiments that allowed for combined measurement of dopamine release with invasive neurophysiology have provided first evidence that beta band oscillations in healthy non-human primates can differentially link dopamine release, beta oscillations and reward and motor control, depending on the contextual information and striatal domain" - This is not very clear - not sure what "differentially link" signifies.

I think the fact that this is not easy to understand signifies the complexity that we and the authors of the cited paper from Ann Graybiel’s lab aimed to communicate. In fact, we stayed very close to the phrasing used in their paper to try and avoid confusion (Title: Dopamine and beta-band oscillations differentially link to striatal value and motor control” - https://doi.org/10.1126/sciadv.abb9226). The specific results go beyond the scope of the discussion but are very interesting, so I would be happy if our paper would inspire readers to look it up.

Changes in manuscript: We have now adapted the sentence to “In line with this more complex picture, direct measurement of dopamine concentration in non-human primates revealed specific interactions between dopamine release, beta oscillations, reward value and motor control, depending on contextual information and striatal domain. This shows that the relationship of dopamine and beta activity is not solely associated with either reward or movement and depends on where in the striatum beta activity is recorded.”

"In fact, one could argue that it can be contextualised in a recently described framework of neural reinforcement, that serves to orchestrate the re-entrance and refinement of neural population dynamics for the production of neural trajectories" - this is not clear - for example what is a neural trajectory? What is meant by "re-entrance and refinement"?

A neural trajectory refers to the path that the activity of a neural population takes through a high-dimensional space over time. It can be obtained through multivariate analysis of population activity with dimensionality reduction techniques, such as PCA. The concept of low-dimensional representations of high-dimensional neural activity has gained a lot of attention in computational neuroscience ever since high-channel count recordings of neural population activity have become available (an early and prominent example is Churchland et al., 2012 Nature https://doi.org/10.1038/nature11129 , while a more recent example is Safaie et al., Nature 2023 https://doi.org/10.1038/s41586-023-06714-0). The review we refer to by Rui Costa and colleagues (Athalye, V. R., Carmena, J. M. & Costa, R. M. Neural reinforcement: re-entering and refining neural dynamics leading to desirable outcomes. Curr Opin Neurobiol 60, 145–154 (2020) https://doi.org/10.1016/j.conb.2019.11.023) suggests that dopamine may serve to modulate the likelihood of a specific pattern to emerge and re-enter the cortex – basal ganglia loop, for the “reliable production of neural trajectories driving skillful behavior on-demand”. We believe that this concept could be revolutionary in our understanding of dopaminergic modulation and disoroders and together with colleague Alessia Cavallo have written an invited perspective on this topic (https://doi.org/10.1111/ejn.16222), which may help further clarify the topic.

Changes in manuscript: We realize that this aspect may sound a bit unclear or far away from the data in this manuscript. However, given that we have spent more than a decade thinking about beta oscillations and how they can be conceptualized, we would prefer not to entirely change our points and rather bet on the possibility that the concepts become more widely accepted and well-known. Nevertheless, we have now adapted the text to make this a bit more clear:

“We hypothesise that, this “status quo” hypothesis could be equally or maybe even more adequately posed on the neural level. Namely, it could provide insights to what degree a certain activity pattern or synaptic connection is to be strengthened or weakened, in light of neural learning. We propose that this putative function can be contextualised in a recently described framework of neural reinforcement, that serves to orchestrate the re-entrance and refinement of neural population dynamics for the production of neural trajectories.”

"....after which it was quickly translated to first experimental studies using cortical or subcortical beta signals in human patients44." - reference 44 only deals with the use of subcortical beta, not cortical, in adaptive control.

The reviewer is right, in fact there is no study using motor cortex beta for adaptive DBS yet, but different studies have used different markers (especially gamma) since then.

Changes in manuscript: We have rephrased and added citations accordingly: “This approach, also termed adaptive DBS, was first demonstrated based on cortical beta activity that was used to adapt pallidal DBS in the MPTP non-human primate model of PD43. It was quickly translated to first experimental studies using subcortical beta signals in human patients44, followed by further research using more complex cortical and subcortical sensing setups and biomarker combinations45,46.”

The paragraph headed " Implications for neurotechnology" is quite long and should be condensed and focused. It doesn't seem to support the last sentence, "....targeted interventions that can increase and decrease beta activity, as recently shown through phase specific modulation45 could be utilised to mimic phasic dopamine release as a neuroprosthetic approach to alter neural reinforcement38." - I don't quite follow the logic. The authors have clearly shown that beta-related circuits tend to be those linked to dopamine modulation, and may subserve tasks for which reinforcement learning is an important mechanism. However the logic of how modulation of beta activity can "substitute" for modulation of dopamine isn't clear. That would seem to require that the mechanism by which dopamine produces reinforcement, is via an effect on beta oscillation properties (phase, amplitude, frequency). Is there evidence for this? If so it should be better spelled out.

We realize that this is very speculative at this point. Indeed, we believe that subthalamic DBS can mimic dopaminergic control and in the future there may be new treatment avenues, e.g. using neurochemical using neurochemical interfaces for which beta could be informative to mimic dopamine release but ultimately explaining this would be very complex, so we have removed the sentence. With regard to the remaining text in the section, we considered shortening / condensing but felt that this paragraph is highly relevant for the ongoing development of neurotechnology and therefore decided to only remove the first and last sentences.

Changes in manuscript: We have removed the first and last sentences.

"While the abovementioned prospects are promising we should cautiously consider the limitations of our study." - an unnecessary sentence to start a "limitations" section, its clearly a paragraph about limitations. In general, authors should go thru discussion and reduce verbosity; it is not nearly as well edited as the rest of the paper.

Agreed.

Changes in manuscript: We removed the sentence.

**Reviewer #3 (Public Review):**
Summary:In this paper, Chikermane et al. leverages a large open dataset of intracranial recordings (sEEG or ECoG) to analyze resting state (eyes closed) oscillatory activity from a variety of human brain areas. The authors identify a dominant proportion of channels in which beta band activity (12-30Hz) is most prominent and subsequently seek to relate this to anatomical connectivity data by using the sEEG/ECoG electrodes as seeds in a large set of MRI data from the human connectome project. This reveals separate regions and white matter tracts for alpha (primarily occipital) and beta (prefrontal cortex and basal ganglia) oscillations. Finally, using a third available dataset of PET imaging, the authors relate the parcellated signals to dopamine signaling as estimated by spatial uptake patterns of dopamine, and reveal a significant correlation between the functional connectivity maps and the dopamine reuptake maps, suggesting a functional relationship between the two.Strengths:Overall, I found the paper well justified, focused on an important topic, and interesting. The authors' use of 3 different open datasets was creative and informative, and it significantly adds to our understanding of different oscillatory networks in the human brain, and their more elusive relation with neuromodulator signaling networks by adding to our knowledge of the association between beta oscillations and dopamine signaling. Even my main comments about the lack of a theta network analysis and discussion points are relatively minor, and I believe this paper is valuable and informative.Weaknesses:The analyses were adequate, and the authors cleverly leveraged these different datasets to build an interesting story. The main aspect I found missing (in addition to some discussion items, see below) was an examination of the theta network. Theta oscillations have been involved in a number of cognitive processes including spatial navigation and memory, and have been proposed to have different potential originating brain regions, and it would be informative to see how their anatomical networks (e.g. as in Figure 2) look like under the author's analyses.The authors devote a significant portion of the discussion to relating their findings to a popular hypothesis for the function of beta oscillations, the maintenance of the "status quo", mostly in the context of motor control. As the authors acknowledge, given the static nature of the data and lack of behavior, this interpretation remains largely speculative and I found it a bit too far-reaching given the data shown in the paper. In contrast, I missed a more detailed discussion on the growing literature indicating a role for beta in mood (e.g. in Kirkby et al. 2018), especially given the apparent lack of hippocampal and amygdala involvement in the paper, which was surprising.

We thank the reviewer for their insightful review of our manuscript. One of the aims of our paper was to provide the ground for a circuit-based conceptualization of beta activity, which does not primarily relate to behavior. Practically we have the ambition to provide a generalizable concept that can be applied to all behavioral domains including mood. The reason we focus on the “status quo” hypothesis, is that it is one of the very few if not only generalizable concept of the function of beta oscillations. Through our paper and the discussion, we have to redirect this concept towards a less cognitive/behavioral and more anatomical network based domain, while acknowledging principles that may overlap. We realize that this is very ambitious and this endeavour is necessarily very complex and not easy to communicate. In light of the reviewers comments, we have made an effort to improve the discussion as best we could without trailing too far away from what our initial aim was. We are thankful for the suggested reference, which we have now added to the discussion in the section where we have previously discussed beta as biomarker for mood, also noting the absence of beta dominant channels in amygdala and hippocampus. Here it should be clarified however, that (a) only three channels were located in the amygdala of which one exhibited beta activity, we should be cautious to not overinterpret this result and (b) most channels exhibited beta and just because beta wasn’t dominant, it doesn’t mean that beta is not present or important in these brain areas. Absence of evidence is not evidence for absence with the way we approached the analysis. We are thankful for the interesting reference, which we have now included our discussion. Notably the study used a complex network analysis, which we could not perform because we did not have parallel recordings from these areas in multiple patients. This is now noted in the limitations.

Changes in manuscript: “For example, it was shown that beta is implicated in working memory28, utilisation of salient sensory cues29, language processing30, motivation31, sleep32, emotion recognition33, mood34 and may even serve as a biomarker for depressive symptom severity in the anterior cingulate cortex35” and “One impactful study reported that beta oscillatory sub-networks of Amygdala and hippocampus could reflect human variations in mood 34. This is interesting, but highlights another relevant limitation of our study, namely that recordings in different areas were stemming from different patients and thus, such sub-network analyses on the oscillatory level could not be conducted.”

Major comment:• Although the proportion of electrodes with theta-dominant oscillations was lower (~15%) than alpha (~22%) or beta (~57%), it would be very valuable to also see the same analyses the authors carried out in these frequency bands extended to theta oscillations.

We agree with the reviewer and appreciate the interest in other frequency bands; theta, alpha and gamma. Our primary interest was to provide a network concept of beta activity, but anticipated that interest would go beyond that frequency band. However, we also had to limit ourselves to what is communicable and comprehensible. The key aim for us was to provide a data-driven circuit description of beta activity that can lay ground for a generalizable concept of where beta oscillations emerge. Reproducing all analyses for every frequency band would clutter both the results and the discussion. Moreover, the honest truth is that funding and individual career plans of the researchers currently do not allow to allocate time for a reanalysis of all data which would be a significant effort. Therefore, we have decided to just add the topography of theta and gamma channels as a supplement. In case the reviewer is interested on a collaboration on extending this project to other frequency bands and circuits, we would like to invite them to get in touch and perhaps this could be a new collaborative project. Until then, we have extended our limitation that this would be important work for the future.

Changes in manuscript:

We have added and cited the new supplementary figure for the results from theta in the results section, which now reads:

“Further information on the topography of theta channels are shown in supplementary figure 1.”

We would like to add that a sensible interpretation of results from gamma dominant channels is unlikely to be possible given the low count of channels with prominent resting activity in this frequency band. We have added the following text to the limitations section: “The aim of this study was to elucidate the circuit architecture of beta oscillations, which is why insights from this study for other frequency bands are limited. Future research investigating the specific circuits of theta, alpha and gamma oscillations and their relationship with neurotransmitter uptake could yield new important insights on the networks underlying human brain rhythms.“

**Reviewer #3 (Recommendations For The Authors):**
Minor comments:• Results: "we performed non-parametric Spearman's correlations between the structural and functional connectivity maps of beta networks with neurotransmitter uptake". This is a significantly complex analysis that requires more detail for the reader to evaluate. There is more detail in the Figure 3 legend but still insufficient. The Methods offer more detail, but I found the description of the parcellation to be vague and I would appreciate a more detailed description.

We thank the reviewer for bringing the insufficient explanation of the methods used to calculate the correlations in analysis to our attention. We have now made an effort to provide more level of detail in the relevant paragraphs.

Changes in manuscript: We have now made changes to both the Results and Methods sections and added the following explanations respectively:

Results: “Next, we resliced the beta network map and the PET images to allow for a meaningful comparison, using a combined parcellation with 476 brain regions that include cortex19, basal ganglia20, and cerebellum21. Here, each parcel – which was a collection of voxels belonging to a particular brain region – from the connectivity map was correlated with the same parcel containing average neurotransmitter uptake from the respective PET scan (see Figure 3A). In this way nonparametric Spearman’s correlations between PET intensity and structural and functional connectivity maps of beta networks were obtained, which indicate to what degree the spatial distribution of connectivity is similar to the distribution of neurotransmitter uptake.“

Methods: “A custom master parcellation in MNI space was created in Matlab using SPM functions by combining three existing parcellations to include cortical regions19, structures of the basal ganglia20 and cerebellar regions21. Regions that were (partially) overlapping between the atlases were only selected once. The final compound parcellation had 476 regions in total. This parcellation was applied to both PET and structural and functional connectivity maps using SPM and custom code. This allowed for the calculation of spatial correlations, providing a statistical measure of spatial similarity of the PET intensity and MRI connectivity distributions. For this, Spearman’s ranked correlations were used to calculate correlations between the PET images, such as the dopamine aggregate map and both functional and structural beta connectivity networks (Figure 3). The analysis was repeated for individual tracers showing similar results Supplementary figure 2. Finally, to validate these results, a control analysis was performed using a GABA PET scan from the same open dataset of neurotransmitter uptake following the same pipeline (Figure 2A, 2B).”

• All of the recordings were taken in an eyes-closed condition. This is likely to affect the power of alpha oscillations; the authors should comment on this.

We agree with the reviewer that this will likely have influenced the results. However, given that the key result of our paper is the abundance and circuit topography of beta oscillations, it is unlikely that increased alpha in some channels will have led to false positive results for beta. If anything, it may have increased the contrast leading to a more conservative estimate of which channels truly show strong beta dominance. On the other hand, we should acknowledge that this limitation can affect the interpretation of the alpha result. Another reason for us to primarily focus on beta in the discussion and results presentation.

Changes in manuscript: We now comment on this in the results:

“It should be noted that that alpha recordings were performed in eyes closed which is known to increase alpha power, which may influence the generalizability of the alpha maps to an eyes open condition. However, given that our primary use of alpha was to act as a control, we believe that this should not affect the interpretability of the key findings of our study.”

• Although the relative proportion of theta and gamma channels is lower, it would be interesting to see the distribution of channels in a SOM figure.

As described above, we have now added supplementary figure 1 that accommodates the topography but not the network analyses.

• Figure legend - typo - "Neither, alpha nor beta" - no comma needed.

Now fixed, thank you for pointing is to this lapse!

• Results: " ere, we aimed to investigate the whole brain circuit representation of beta activity, which is impossible with current neurophysiology approaches" not entirely accurate; suggest rephrasing it to "Here, we aimed to investigate the whole brain circuit representation of beta activity, which is impossible with non-invasive neurophysiology approaches "

Thank you for suggesting the alternative formulation.

Changes in manuscript: The text has been modified as per the suggestion and now reads “Here, we aimed to investigate the whole brain circuit representation of beta activity, which is impossible with non-invasive neurophysiology approaches”.

• Results - typo - "cortical brain areas, that exhibit resting beta activity share a common brain network" - no comma needed.

Thank you for the suggestion, the comma has been removed to better the flow of the sentence structure as suggested.